# Mıko: Multimodal Intention Knowledge Distillation from Large Language Models for Social-Media Commonsense Discovery

Feihong Lu*
Beihang University
Beijing, China
lufh@act.buaa.edu.cn

Weiqi Wang*
Hong Kong University of Science and
Technology
Hong Kong SAR, China
wwangbw@cse.ust.hk

Yangyifei Luo
Beihang University
Beijing, China
luoyangyifei@buaa.edu.cn

Ziqin Zhu
Beihang University
Beijing, China

Qingyun Sun†
Beihang University
Beijing, China
sunqy@buaa.edu.cn

Baixuan Xu
Hong Kong University of Science and
Technology
Hong Kong SAR, China

Haochen Shi
Hong Kong University of Science and
Technology
Hong Kong SAR, China

Shiqi Gao
Beihang University
Beijing, China

Qian Li
Beijing University of Posts and
Telecommunications
Beijing, China

Yangqiu Song
Hong Kong University of Science and
Technology
Hong Kong SAR, China

Jianxin Li
Beihang University
Beijing, China

## Abstract

Social media has become ubiquitous for connecting with others, staying updated with news, expressing opinions, and finding entertainment. However, understanding the intention behind social media posts remains challenging due to the implicit and commonsense nature of these intentions, the need for cross-modality understanding of both text and images, and the presence of noisy information such as hashtags, misspelled words, and complicated abbreviations. To address these challenges, we present Mıko , a **M**ultimodal **I**ntention **K**nowledge Distillati**O**n framework that collaboratively leverages a Large Language Model (LLM) and a Multimodal Large Language Model (MLLM) to uncover users' intentions. Specifically, our approach uses an MLLM to interpret the image, an LLM to extract key information from the text, and another LLM to generate intentions. By applying Mıko to publicly available social media datasets, we construct an intention knowledge base featuring 1,372K intentions rooted in 137,287 posts. Moreover, We conduct a two-stage annotation to verify the quality of the generated knowledge and benchmark the performance of widely used

LLMs for intention generation, and further apply Mıko to a sarcasm detection dataset and distill a student model to demonstrate the downstream benefits of applying intention knowledge.

## CCS Concepts

• **Computing methodologies → Knowledge representation and reasoning**.

## Keywords

Social Media, Intention Knowledge Distillation, Large Language Model, Large Vision Language Model

**ACM Reference Format:**
Feihong Lu, Weiqi Wang, Yangyifei Luo, Ziqin Zhu, Qingyun Sun, Baixuan Xu, Haochen Shi, Shiqi Gao, Qian Li, Yangqiu Song, and Jianxin Li. 2024. Mıko: Multimodal Intention Knowledge Distillation from Large Language Models for Social-Media Commonsense Discovery. In *Proceedings of the 32nd ACM International Conference on Multimedia (MM '24), October 28-November 1, 2024, Melbourne, VIC, Australia.* ACM, New York, NY, USA, 10 pages. https://doi.org/10.1145/3664647.3681339

*Both authors contributed equally to this work.
†Qingyun Sun is the corresponding author.

## 1 Introduction

Social media platforms serve as a cornerstone in our daily lives, which trigger various data mining and Natural Language Processing (NLP) tasks that require deep understanding of users' behaviors [4, 18, 28, 53]. However, according to psychological theories [5, 52], the interrelation of intention reflecting human motivation significantly influences behavioral patterns. Intentions are mental states or processes of planning, directing, or aiming towards a desired outcome or goal [6]. It is widely acknowledged in scholarly discourse that intention is interwoven with a form of desire,

thereby rendering intentional behavior as inherently valuable or desirable [73], which makes it an irreplaceable component for agents with the theory of mind [3, 11, 22]. For example, in Figure 1, users' intentions are strongly correlated to the contents of their social media posts. Thus, in social media, accurately understanding users' intentions in their posts has the potential to motivate downstream tasks as they provide a more cognitively shaped observation of the posts. In recent years, there has been a surge in the development and enhancement of intention discovery algorithms, with applications spanning various fields such as sentiment analysis [76], online shopping [17, 68, 71, 72] with conceptualizations [30, 63–65], and social good [1, 21], which aim to improve the performance of downstream tasks by gaining insights into user intentions. Given the existence of the "dark side" of social media, characterized by the dissemination of harmful content [2, 15], the analysis of social media content to discern underlying motives and intentions is an imperative and pressing issue.

However, identifying users' intentions in large-scale social media platforms remains nontrivial. Several challenges stand out throughout this process. First, intentions in the text are often implied rather than explicitly stated, which makes it impossible for heuristically or semantically designed extraction methods to retrieve from open-domain data. Furthermore, social media data's inherently multimodal nature, which encompasses a rich tapestry of textual, visual, and auditory elements, significantly magnifies this complexity. This diversity in user-generated content demands more advanced and nuanced methods of analysis. Last but not least, the prevalent presence of "noise" in social media posts, including hashtags, misspelled words, and complex abbreviations, poses substantial interpretative challenges for existing analytical models. Despite ongoing research efforts, there remains a discernible gap in methodologies for social intention analysis, particularly within the context of social media. As a result, our research is primarily motivated by the exploration of automated techniques for identifying multimodal social intentions within open domains.

Owing to the abundant knowledge and robust reasoning abilities of Large Language Models (LLMs) [10, 43, 47, 48, 58, 59, 75], an increasing number of researchers have shown their superior performances on various tasks [13, 25, 37], such as product recommendation [45], sentiment analysis [60], and mental health analysis [70]. However, several concerns exist when leveraging them to reveal the intentions of social media posts. First, content generated by LLMs, especially when solely relying on social media posts, can be unreliable. They may generate hallucinatory outputs, such as the generation of uncontrollable, inaccurate content, and the misinterpretation of irrelevant input information. Moreover, social media posts often comprise both textual and visual elements, necessitating an in-depth understanding of each modality and the ability to perform cross-modal reasoning. For instance, as depicted in Figure 1, user 2 intends to express dissatisfaction and anger with the Lakers' recent performance. This requires a combined understanding of both text and image in the post to accurately analyze the user's intention.

To tackle all issues above, in this paper, we present **Miko**, a **M**ultimodal **I**ntention **K**nowledge Distillati**O**n framework, to acquire intention knowledge based on large-scale social media data.

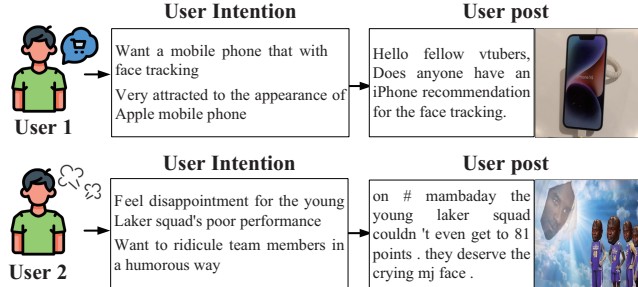

**Figure 1: Examples of users' intentions in their social media posts. User 1's intention is to buy a cost-effective iPhone, while User 2's intention is to be disappointed with the performance of the young Lakers players.**

Specifically, Miko originates from analyzing extensive user behaviors indicative of sustainable intentions, such as various posting activities. Given a social media post and its accompanying image, we use a Multimodal Large Language Model (MLLM) to generate descriptions of the input images based on the textual content of the post. Following this, we instruct a large language model (LLM) to extract key information from both the input text and image descriptions to minimize the impact of noisy information in text. After both processing steps, we finally instruct a powerful LLM, such as Chat-GPT [47], to generate potential intentions underlying these posting behaviors as viable candidates. We align our prompts with 9 specific commonsense relations in ATOMIC [54], a popular commonsense knowledge base for social interactions, to make the intentions comprehensive in a commonsense manner. Another open-prompted relation is also used to maintain knowledge diversity.

We evaluate Miko from both intrinsic and extrinsic perspectives. Intrinsically, we compile a series of publicly available social media datasets and apply Miko to them to obtain the intentions in their social media posts. A two-stage annotation is then conducted to evaluate the plausibility and typicality of the generated contents. We then leverage intentions with top ratings in annotations as benchmark data to evaluate the capability of other generative LLMs. Experiment results show that (1) Miko is capable of generating intentions that are highly plausible and typical to the user's original post and (2) most LLMs fail at generating high-quality intentions while fine-tuning on Miko generated intentions resolve this issue. Extrinsically, we evaluate the downstream benefits of generated intentions by applying them to a sarcasm detection task, showcasing that incorporating intentions in current methods leads to state-of-the-art performances[1]. In summary, this paper's contributions can be summarized as follows.

- We present Miko, a novel distillation framework designed to automatically obtain intentions behind social media posts with the assistance of LLMs and MLLMs. Miko stands out with its unique design in bridging the gap between understanding text and image in a social media post simultaneously with two large generative models.
- We conduct extensive human annotations to show the superiority of the generated intentions in terms of both plausibility

---

[1]Our code and data are publicly available at https://github.com/RingBDStack/Miko.

and typically. Further experiments show that most large generative models face challenges when prompted to generate intentions directly while fine-tuning them on Miko generated intentions helps significantly.

- We further conduct experiments to show that intentions generated by Miko can benefit the sarcasm detection task, which highlights the importance of distilling intentions in social media understanding tasks.

## 2 Related work

### 2.1 Intentions in Social Media

Intention is closely related to psychological states, such as beliefs and desires [5, 52]. It is generally believed that intention involves some form of desire: the behavior of intention is considered good or desirable in a certain sense [73]. This aspect enables intentions to inspire current human behavior, among which users' posting behavior is a typical behavior driven by intentions. In social media platforms, positive social posts (such as charity, mutual help, etc.) will promote social development and progress, while negative social posts (such as ridicule, abuse, oppositional remarks, etc.) can cause harm to people's hearts and hinder social peace. Recently, the widespread application of social media in daily life has aroused the interest of scholars, which use intentional knowledge to tackle the task of sentiment analysis [29, 76], hate speech detection [66], recommendation system [20, 27, 36] et al. It aims to enhance downstream task performance by leveraging insights into user intentions. Thus, analyzing social media content to discern underlying motives and intentions is an imperative and valuable issue.

In sentiment analysis tasks, understanding user content ideas is crucial, enabling a deeper insight into their emotional states and potential needs. This aspect, as elaborated in the work of Zhou et al. [76], is fundamental for accurately classifying user sentiments. Through meticulous extraction and analysis of contextual clues and posting intentions in user-generated content, sentiment analysis tools significantly enhance their ability to categorize sentiments into well-defined categories such as positive, negative, or neutral. In recommendation systems, existing works often use user repurchase intentions to analyze customer needs and achieve more accurate recommendations. As [31] says the consumer's purchase intention is the propensity of consumers to continue participating in retailers' or suppliers' commercial activities.

However, the task of identifying user intentions within the vast, open-domain web and analyzing the conveyed information presents significant challenges. These challenges stem from the sheer volume of data produced across numerous websites. It is difficult for traditional algorithm models to accurately locate key information and extract the accurate intentions of users. This difficulty stems from the complexity and diversity of user-generated content, which requires more advanced and nuanced analysis methods. We are the first to propose the open-domain social intention generation framework to extract accurate and reasonable social intentions from multimodal social posts.

### 2.2 Knowledge Distillation

Knowledge distillation [32] is a strategy in which a pre-trained model (known as the teacher model) facilitates the training of a

**Table 1: Statistics of the using datasets. "Statistics" refers to the number of posts contained in each dataset, whereas "Sample" denotes a randomly selected subset from each dataset.**

| Dataset | Statistics | Sample | |
|---|---|---|---|
| Twitter2015 | 8,257 | Walking home from school over blood stains in Gresham, hours after a man died. |  |
| Twitter2017 | 4,395 | Incredibly busy at KW Multicultural festival! # lovemyhood@ChiefRehill @MichaelMayKit |  |
| Twitter100k | 100,000 | Hey, @HelloAlfred.You ruined a pair of my shoes. Not cool. Goodyear? |  |
| Twitter Sarcasm | 24,635 | well , we can thank our lucky stars this thing is still standing |  |

secondary model (termed the student model). With the development of Large Language Models (LLMs), more and more researchers are trying to guide and refine domain-specific knowledge from LLMs into small models, thereby enhancing the generalization capabilities of small models [12, 26, 41, 56, 61, 62]. Liu et al. [42] attempts to distill time series information from LLMs into small models, where the student network is trained to mimic the features of the LLM-based teacher network that is pre-trained on large-scale datasets. Sun et al. [55] design an effective and efficient Pre-trained Recommendation Models (PRM) distillation framework in the multi-domain recommendation to accelerate the practical usage of PRMs.

However, the above-mentioned studies concentrate on extracting direct information from large language models (LLMs) but overlook a hierarchical analysis to identify pertinent information. They are primarily applied in specific fields without analyzing the motives or intentions of social users. Our framework, referred to as Miko, can be seen as the first attempt to utilize LLMs for the distillation and analysis of social intentions.

## 3 Definitions and Datasets

### 3.1 Task Definitions

In the context of analyzing a post, denoted as $t$, and its accompanying image as $m$, the objective of the intention knowledge distillation task is to extract a set of intentions, represented as $k$, from both post $t$ and image $m$. Aligning with most of the current research in intention analysis, this task is approached as an open domain generation problem. Let $t = (t_1, t_2, ..., t_n)$ symbolize a sequence of input words in the post, $k = (k_1, k_2, ..., k_l)$ represents the set of intentions that are deduced from various aspects of both the post text and the image, where $n$ and $l$ indicate the length of the post and the top-$l$ most relevant intentions, respectively.

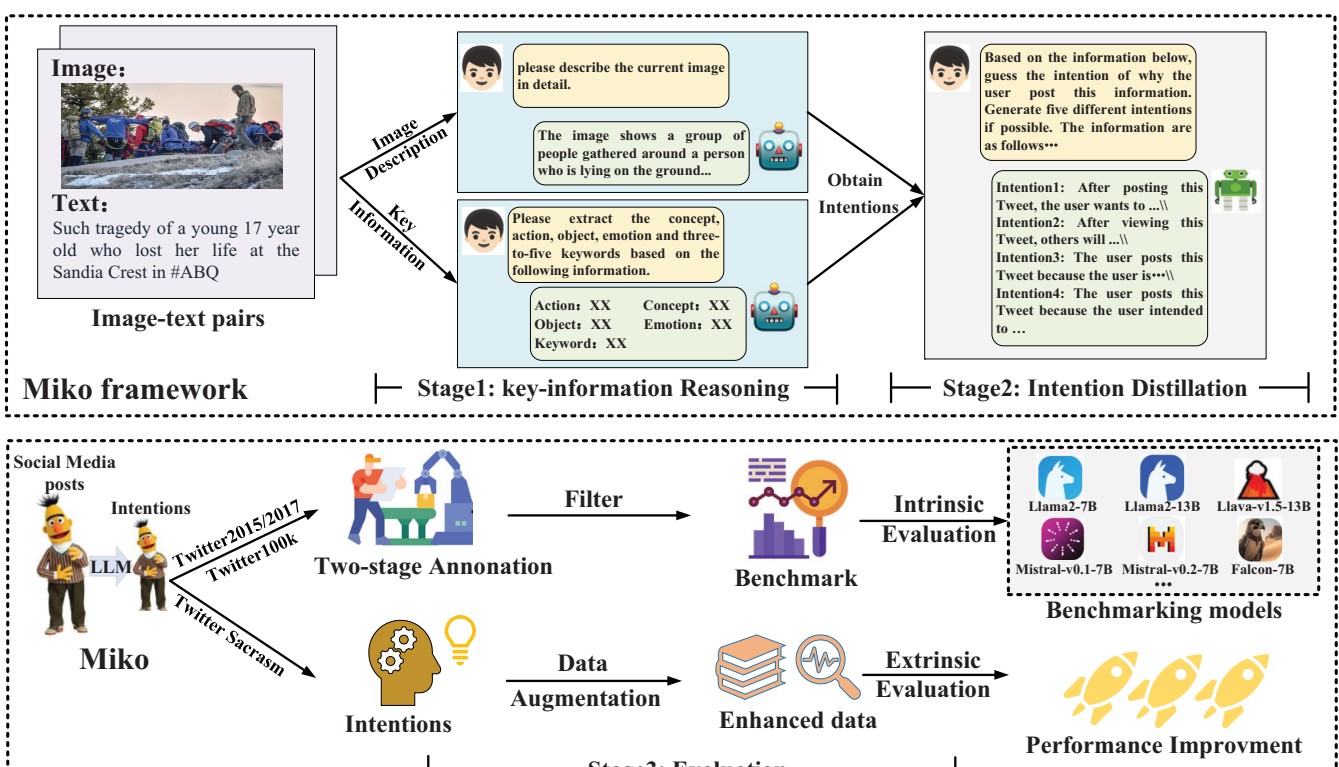

**Figure 2: The overall architecture of our work, which encompasses three core components: multi-information reasoning, intention distillation, and multi-view intention effectiveness evaluation. We leverage the LLava and ChatGPT models, employing a novel hierarchical prompt guidance approach to extract image description (*Section* 4.1), key information (*Section* 4.2) and intentions (*Section* 4.3) from user posts. Following this, we annotate the derived intentions based on rationality and credibility, create a benchmark (*Section* 5.1), and assess the performance of various LLMs (*Section* 5.3) and the performance with the help of intentions on sarcasm detection task (*Section* 5.4).**

## 3.2 Datasets

On the task of intention generation, we utilize four renowned public datasets to address the challenges posed by the diversity of social media posts, providing a more robust and comprehensive analysis of social media interactions. The datasets include Twitter-2015 [74], Twitter-2017 [46], Twitter100k [33], and Twitter Sarcasm [7]. The datasets comprise 8,357 sentences for Twitter 2015, 4,819 sentences for Twitter 2017, 100,000 sentences for Twitter100k, and 24,635 sentences for Twitter Sarcasm. Statistics are shown in Table 1.

## 4 Method

In this section, we present Mɪᴋᴏ, a Multimodal Intention Knowledge Distillation framework, which is shown in Figure 2. Mɪᴋᴏ can mainly be summarized in three steps. Given an image and text pair in a social media post, we start by instructing an MLLM to generate the descriptions of images in social media posts in natural language form to bridge vision and text modalities. Then an LLM is simultaneously instructed to analyze the text in each post by extracting key information according to five pre-defined key dimensions. Utilizing the extracted middle-step information, we finally instruct the LLM again to let it generate the underlying intentions of users' posts and construct multi-perspective intention profiles.

## 4.1 Image Captioning

When users post, the images attached to the posts often contain their potential posting motivations, which are mainly reflected in two aspects. First of all, when the images cannot be directly expressed in text form, such as sarcastic remarks, it is usually because the content that users want to express may violate the speech restrictions of the platform. In this case, images become an alternative means of expression, allowing users to bypass the limitations of text and convey their true intentions. Secondly, users may use images to further explain or strengthen the message of the text, making the original post content richer and clearer. Such images not only supplement the text, but in many cases they help the public understand the intention and emotion behind the post more deeply and accurately. To this end, we utilize the advanced Multimodal Large Language Model, LLava[43] for image captioning, the accessed code is `liuhaotian/LLaVa-v1.5-13b`. With the help of a special design prompt, LLava is utilized to derive detailed descriptions of image information from the raw image-text pairs. This approach ensures a richer and more nuanced interpretation of the social media post. The structured prompt we employ is as follows:

*Based on the following text "<Post text information>", please describe the current image in detail.*

*Note that for the input of single text information, we do not perform this step of processing.*

## 4.2    Chain-of-Key Information Reasoning

Social media posts frequently contain noisy elements like hashtags, misspellings, and complex abbreviations, which could influence the performance of intention analysis. In addition, since LLM faces difficulties in accurately describing and extracting useful information in the original posts, which may lead to hallucinations, it is necessary to further extract more crucial information from both the original post and the corresponding image descriptions after obtaining the descriptions of the images to eliminate the influence of noise information. We design a key information prompting strategy to guide ChatGPT [47] in obtaining the concept, action, object, emotion, and keywords from different dimensions of the original post. For ChatGPT, we access it through Microsoft Azure APIs. The code for the accessed version of ChatGPT is gpt-35-turbo (access version 2024-02-01). The structured prompt we employ is as follows:

*Please extract the concept, action, object, emotion, and three to five keywords based on the following information.*

*Note: remove the person's name and other information, retain only the key information. The information is <Text information>/ <Image description>.*

## 4.3    Intention Distillation

Employing LLMs directly to extract users' posting intentions can lead to challenges, including superficial comprehension and inaccurate understanding. To mitigate these issues and improve the capacity of models to accurately and fully grasp the intentions behind social media posts, we have developed an intention distillation strategy, which combines the original post information, image description information, and key information to generate a more accurate and comprehensive open-domain and standardized description of the original posting intention.

In addition, users' posting intentions are open and diverse. Therefore, to comprehensively and accurately analyze users' posting intentions from multiple perspectives, we further refined and standardized the categorization of the generated intentions with nine types of relations defined in ATOMIC [54], including "xNeed" (user's need), "xIntent" (user's intention), "xAttr" (user's attribute), "xEffect" (effect of user's action), "xReact" (user's reaction), "xWant" (user's desire), "oEffect" (impact on others), "oReact" (others' reaction), and "oWant" (others' desire), along with an open intention termed "Open". Here, "x" represents the thoughts and behaviors of the user after posting, while "o" denotes the impact of the post on others. "Open", as an open-domain intention, describes the motive and purpose behind a user's decision to publish a specific post content. By employing this method of intention categorization, we can comprehensively analyze users' posting intentions and accurately grasp the motives behind user posts, thereby deepening our understanding of user behavior. The specific prompt design is included in the supplementary materials. In this step, we also use ChatGPT gpt-35-turbo (access version 2024-02-01) to obtain user intentions.

## 5    Intrinsic Evaluations

In this study, we conducted intrinsic evaluations of the generated intentions. To assess the quality of intention generation, we randomly selected 1,000 posts with 10,000 all aspects of intentions and performed manual annotations in *Section* 5.1, and finally, we extracted accurate and comprehensive intents consistent with human logic and added them to the benchmark. Furthermore, we conducted a comprehensive evaluation of intentions generated by the Mɪᴋᴏ framework, encompassing aspects such as knowledge quality case study (*Section* 5.3). Subsequently, based on the intentions obtained in *Section* 4.3, we trained a local LLM model (*Section* 5.2) and used the benchmark to evaluate the performance of other LLMs in generating intentions, as well as the intention generation performance of the trained model (*Section* 5.4).

## 5.1    Two-stage Annotation

As the generated intentions can be incorrect or not rational, refer to the approach of FolkScope [72], we apply the human annotation to obtain high-quality assertions and then to determine the rationality of the intention generation, which as a benchmark for evaluating the ability of generating intentions when using other models. We use Label Studio [57] to annotate the intention data. In this stage, five annotators are provided with generated candidates' intentions and raw text-image pairs.

To acquire high-quality intention data as a benchmark for evaluating other models, our initial strategy involves assessing their typicality. We randomly selected 1,000 Twitter posts along with their respective intention data from our dataset for human assessment. As detailed in *Section* 4.3, each post encompasses 10 distinct types of intention data. We evaluate the intention information of each post individually, assigning scores based on the following criteria: "1 point for a high typicality", "0 points for a low typicality", and "-1 point for a implausible".

Following the evaluation of the generated intentions' typicality, it is crucial to assess whether each annotated post needs to be added to the benchmark. This step ensures that the annotations are not only rational but also objective. Moving beyond the basic typicality judgments for aspects' of the intention, our second step introduces more nuanced and precise measures of typicality, focusing on informativeness and comprehensiveness. In this phase, we conduct a statistical analysis of the data results marked in the previous step and calculate the total score of different generated intents for each post. For posts with a total score exceeding 5, we further conduct discrimination manually. Ultimately, we retain those that conform to human logic and possess comprehensive intention information, adding them to the benchmark. This serves as a basis for evaluating other knowledge distillation and intent generation methods. The number of posting intentions from different perspectives is shown in Table 2.

## 5.2    Distillation Evaluations

*5.2.1    Knowledge Quality.* The primary objective of the Knowledge Quality Evaluation is to identify and recognize high-quality knowledge accurately. In this context, we focus on assessing whether the intentions generated are of superior quality. For this purpose,

| Social Media Post | Key-information | Intentions | P | T |
|---|---|---|---|---|
| Firefighter killed batting plane fire at Dubai airport. The airport has resumed all flights | **Image description:** The image shows a large airplane on a runway at an airport, with smoke billowing from it. The airplane appears to be on fire, and a firefighter is present, trying to extinguish the flames. The scene suggests that there has been an incident at the airport, possibly involving the airplane. Despite the ongoing emergency, the airport has resumed all flights, indicating that the situation is being managed and controlled

**Concept:** Airport emergency response.
**Action:** Battling a plane fire.
**Object:** Airplane.
**Emotion:** Tragedy.
**Keywords:** Firefighter, Dubai airport, smoke, flames, resumed flights. | **xWant:** After posting this Tweet, the user wants to inform their followers about the tragic incident at Dubai airport. | H | H |
|  |  | **oEffect:** After viewing this Tweet, others will be updated on the situation at Dubai airport and be aware of any potential delays or cancellations. | H | H |
|  |  | **xAttr:** The user posts this Tweet because the user is concerned about the safety of the passengers and airport personnel at Dubai airport. | H | H |
|  |  | **xIntent:** The user posts this Tweet because the user intended to raise awareness about the importance of emergency response teams at airports. | H | L |
|  |  | **xReact:** After posting this Tweet, the user feels a sense of responsibility to inform the public about the incident. | H | H |
|  |  | **oReact:** After viewing this Tweet, others feel a sense of concern for the safety of those at Dubai airport. | H | H |
|  |  | **oWant:** After viewing this Tweet, others want to know more about the incident and whether any measures are being taken to prevent future emergencies. | H | H |
|  |  | **oEffect:** After posting this Tweet, the user will continue to monitor the situation at Dubai airport and update their followers if needed. | L | L |
|  |  | **xNeed:** Before posting this Tweet, the user needs to verify the accuracy of the information and ensure that it is not causing any panic or unnecessary alarm. | H | H |
|  |  | **Open:** The user posted this tweet because they want to pay tribute to the brave firefighter who lost their life while trying to save others. | L | L |

**Figure 3: An example illustrates the generated image description, key information and intentions. "P" stands for the plausibility and "T" stands for the typicality. Generated tails with good quality (in green) and bad quality (in red) are highlighted. Besides, "H" and "L" indicates the high and low plausibility and typicality scores respectively.**

**Table 2: Statistics on the number of used intentions in the benchmark we constructed.**

| Relation | xWant | oEffect | xAttr | xIntent | xReact | oReact | oWant | oEffect | xNeed | Open | Average | Total |
|---|---|---|---|---|---|---|---|---|---|---|---|---|
| Numbers | 853 | 837 | 799 | 818 | 654 | 772 | 828 | 758 | 717 | 832 | 787 | 7,868 |

we conducted a human evaluation on the results of two-stage annotation in *Section* 5.1, as shown in Figure 4. It can be seen that on the 10 different aspects of generated intentions, most of the results generated by our framework are "high typicality"(more than 80%). Only at least a few samples are "low typology" (not exceeding 10%) and "implausible" (not exceeding 10%), which is evident that most instances generated by the MIKO framework demonstrate a high degree of correlation with human cognition. This finding means the intention information produced by MIKO largely aligns with the process and manner of human cognition and thinking, which involves initially identifying key information from raw data, followed by conducting a more in-depth analysis of the original content under the guidance of this key information. However, it is noteworthy that, despite the high quality of most intentions, certain categories of intention, such as "xReact", show some deviation from human understanding. This suggests that even LLMs struggle to fully comprehend users' feelings and perceptions, marking an important area for future research.

*5.2.2 Case Study.* We show an example of a raw text-image pair and their corresponding knowledge as well as image descriptions (*Section* 4.1), key information (*Section* 4.2), and different aspects of generated intentions (*Section* 4.3) in Figure 3. We use plausibility and typicality to measure the quality of generated information, which can observe that the majority of the generated intentions are both reasonable and comprehensive, aligning with human intuitive understanding. For instance, intentions like "After posting this Tweet, the user aims to inform their followers about the tragic

incident at Dubai airport" and "Upon viewing this Tweet, others will be updated on the situation at Dubai airport and become aware of any potential delays or cancellations" are examples of such. As a result, some of the open intentions are very good as well, and only a very small number of examples generate low quality.

## 5.3 Benckmark Other LLMs

We are interested in whether using different types of language models without using the MIKO framework has a significant impact on the generated intention. Hence we empirically analyze the plausible rate of generation using eleven LLMs: LLama2-7B [59], LLama2-13B [59], Mistral-7B-Instruct-v0.1 [34], Mistral-7B-Instruct-v0.2 [34], Falcon-7B [50], Flan-T5-xxl-11B [14], GLM3 [19], GLM4 [19], LLava-v1.5-13B [43] and LLava-v1.6-vicuna-7B [43].

Besides, to enhance the efficacy of intention generation from social posts using a locally deployed model, we leverage the LLama2-7B as its effectiveness has been demonstrated in several open-source language-only instruction-tuning works [23, 51]. The LLama2-7B model's selection was motivated by its balance of computational efficiency and linguistic capability, making it a pragmatic choice for local deployment scenarios where resource constraints are considered. At this stage, the user intentions $k$ identified in *Section* 4.3 are leveraged to craft instruction pairs to instruction finetune the LLM. In detail, for each post $t$ and associated image $m$, we utilize the LLM to furnish the image description $X_v = g(m)$ and key information $X_i = g(t, m)$, where $g(\cdot)$ represents the text generated by the LLM. Subsequently, we formulate the training instruction

**Table 3: Average BERTscore (reported as percentages) for the 10 different aspects of the generated intentions. Note that the results presented here have been adjusted to exclude prefixes such as "After posting this Tweet, the user wants to." "LoRA Fine-tuned" indicates a model trained using intentions via instruction finetuning. "*" indicates that image descriptions and key information were not used.**

| Model | xWant | oEffect | xAttr | xIntent | xReact | oReact | oWant | xEffect | xNeed | Open | Average |
|---|---|---|---|---|---|---|---|---|---|---|---|
| LLama2-7B | 62.13 | 60.05 | 57.39 | 57.61 | 54.27 | 55.99 | 58.73 | 53.26 | 58.92 | 54.17 | 57.25 |
| LLama2-13B | 62.51 | 59.72 | 57.27 | 55.96 | 56.00 | 54.33 | 56.94 | 52.28 | 59.49 | 52.20 | 56.67 |
| Mistral-7B-Instruct-v0.1 | 63.06 | 60.51 | 56.48 | 57.99 | 52.83 | 57.12 | 60.58 | 52.91 | 57.85 | 53.18 | 57.25 |
| Mistral-7B-Instruct-v0.2 | 61.47 | 59.97 | 55.85 | 58.94 | 54.76 | 56.40 | 57.90 | 54.55 | 58.40 | 53.15 | 57.14 |
| Falcon-7B | 63.97 | 58.66 | 58.01 | 56.79 | 55.19 | 57.09 | 57.21 | 52.35 | 57.10 | 53.91 | 57.03 |
| Flan-T5-xxl-11B | 63.53 | 60.25 | 55.46 | 57.03 | 53.01 | 56.97 | 56.86 | 51.61 | 57.97 | 54.78 | 56.75 |
| GLM3 | 66.09 | 59.99 | 60.44 | 58.16 | 57.87 | 58.61 | 59.09 | 58.17 | 57.89 | 67.83 | 60.41 |
| GLM4 | 64.76 | 59.33 | 57.17 | 52.84 | 53.82 | 53.79 | 56.87 | 56.13 | 54.77 | 65.56 | 57.50 |
| LLava-v1.5-13B | 69.24 | 62.79 | 56.00 | 50.99 | 57.40 | 59.31 | 61.05 | 61.98 | 57.32 | 69.67 | 60.58 |
| LLava-v1.6-vicuna-7B | 67.66 | 61.14 | 63.03 | 56.50 | 58.03 | 58.51 | 60.72 | 56.17 | 58.91 | **69.87** | 61.05 |
| LLama2-7B (LoRA Fine-tuned*) | 64.06 | 56.92 | 64.05 | 57.63 | 60.10 | 59.31 | 59.72 | 58.61 | 56.60 | 48.59 | 58.60 |
| LLama2-7B (LoRA Fine-tuned) | **69.60** | **64.89** | **66.56** | **61.39** | **62.25** | **62.45** | **63.08** | **62.44** | **60.23** | 57.67 | **63.06** |

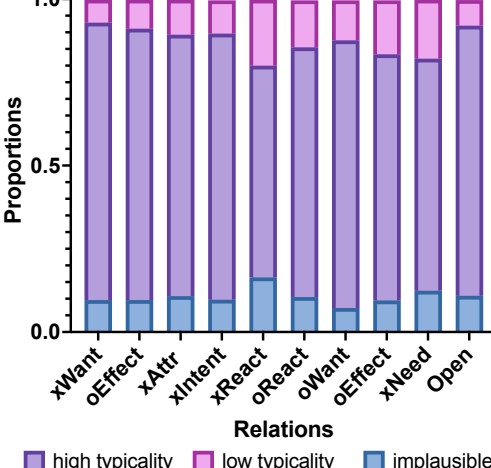

**Figure 4: Average typicality score of each aspect of intentions. The vertical axis represents the proportion of three different categories within manually annotated intentions, while the horizontal axis displays ten different aspects of intentions.**

$X_{instruct}^w = (t_q^1, t_a^1, \ldots, t_q^N, t_a^N)$ for each post, where $N$ denotes the total number of intentions for the post, $t_q$ signifies the outcome derived from integrating the $t$, $X_v$, $X_i$, and specific intent-generating prompts. These intentions are arranged sequentially, with all answers treated as responses from the assistant.

As shown in Table 3, it is observed that the multimodal large models outperform text-based LLMs such as LLama2-7B, GLM3, and GLM4. This suggests that the inclusion of image information in user posts may reveal latent purposes and psychological activities, thereby enabling the model to analyze and identify users' posting intentions more accurately. Furthermore, training the LLama2-7B model with distilled intention knowledge significantly enhances its capability in intention analysis, underscoring the effectiveness

and validity of our extracted intention knowledge in guiding the model's extraction of intention knowledge accurately. Moreover, an intriguing observation is made: the performance of GLM4 is inferior to that of GLM3. This discrepancy is hypothesized to be due to GLM4's training on a substantially larger dataset of Chinese language materials, which may result in its reduced proficiency in interpreting English social media posts compared to GLM3.

## 6 Extrinsic Evaluation

To further validate the effectiveness of the generated intentions and their ability to enhance the accuracy of downstream tasks, we have appended the generated intentions to the sarcasm detection task and conducted an evaluation. For the original image-text data in the sarcasm detection data, we initially apply the prompt design from *Section* 4.1 to obtain descriptions of the current input images and to distill the key information (*Section* 4.2) contained in the image-text pairs. Subsequently, we use the raw texts, image descriptions, and key information as inputs, employing the generated intentions designed in *Section* 4.3 to extract the posting intentions of the users. These intentions are then appended to the raw posts and image descriptions, serving as inputs for training the model and evaluating test data. In this case, we can obtain social intentions that are most relevant to the context in downstream tasks.

### 6.1 Setup

We conducted experiments on the twitter sarcasm dataset, which is collected by [7]. This dataset contains English tweets expressing sarcasm labeled as "1" and those expressing non-sarcasm labeled as "0". For a fair comparison, we meticulously cleaned our dataset, removing instances with missing image modality data. Then, we reproduce our Mɪᴋᴏ framework on the cleaned dataset to obtain image descriptions and intentions of the source data. For knowledge extraction, we employed LLava [43] for extracting image descriptions and leveraged ChatGPT [47] for intentions extraction. To determine if the methodologies inspired by Mɪᴋᴏ genuinely improve

**Table 4: Comparison results for sarcasm detection. "INTE" represents the social intention derived from MIKO, and "IMGDES" refers to the image descriptions generated via LLava."Text" refers to only use raw posts. † indicates ResNet backbone and ‡ indicates ViT backbone. Additionally,**

|  | Model | Acc(%) | P(%) | R(%) | F1(%) |
|---|---|---|---|---|---|
| Text | TextCNN | 80.03 | 74.29 | 76.39 | 75.32 |
|  | Bi-LSTM | 81.90 | 76.66 | 78.42 | 77.53 |
|  | SMSD | 80.90 | 76.46 | 75.18 | 75.82 |
|  | BERT-(Text) | 83.85 | 78.72 | 82.27 | 80.22 |
| Image | ResNet | 64.76 | 54.41 | 70.80 | 61.53 |
|  | ViT | 67.83 | 57.93 | 70.07 | 63.43 |
|  | BERT-(IMGDES) | 75.15 | 67.45 | 72.46 | 69.86 |
| Multimodal | HFM† | 83.44 | 76.57 | 84.15 | 80.18 |
|  | D&R Net† | 84.02 | 77.97 | 83.42 | 80.60 |
|  | Att-BERT† | 86.05 | 80.87 | 85.08 | 82.92 |
|  | InCrossMGs‡ | 86.10 | 81.38 | 84.36 | 82.84 |
|  | CMGCN‡ | 86.54 | – | – | 82.73 |
|  | HKE† | 87.02 | **82.97** | 84.90 | 83.92 |
|  | HKE‡ | **87.36** | 81.84 | 86.48 | 84.09 |
|  | BERT-(Text+IMGDES) | 86.89 | 82.06 | 85.76 | 83.87 |
|  | BERT-(Text+INTE) | 87.14 | 82.43 | 85.97 | 84.16 |
|  | BERT-(Text+IMGDES+INTE) | 87.22 | 82.08 | **86.81** | **84.38** |

sarcasm detection accuracy, we adopted the pre-trained BERT-base-uncased model [16] as the textual backbone network. This setup was used to obtain initial embeddings for texts and knowledge. We then enhanced the original text by appending image descriptions and the extracted intentions. This approach enabled us to assess whether the social intention knowledge extracted by MIKO contributes additional valuable insights to the sarcasm detection task.

## 6.2 Baselines

In our study, referring to the experiments of **HKE** [44], we utilize both text-based and multimodal approaches as baseline frameworks to evaluate the impact of generated intentions referenced by**HKE**. For text-based methods, we integrate **TextCNN** [35], **Bi-LSTM** [24], and **SMSD** [67]. Additionally, we adopt **BERT** [16], a robust baseline in sarcasm detection. In the multimodal domain, our baselines encompass **Image** [8],**ViT** [38], **HFM** [9], **D&R Net** [69], **Att-BERT** [49], **InCrossMGs** [39], a modified version of **CMGCN** [40] that excludes external knowledge, and **HKE** [44], which proposed a hierarchical framework for sarcasm detection.

## 6.3 Results and Analysis

In our preliminary evaluation, we assessed the efficacy of our proposed framework against established baseline models. The corresponding accuracy (Acc), precision (P), recall (R), and F1 score (F1) are shown in Table 4. The outcomes indicate that the BERT model

achieves state-of-the-art performance with the help of intention data. From the Table, we can observe that: 1) Text-based models exhibit superior performance over image-based methods, highlighting that text is easier to interpret and more information-dense than images. This finding confirms the validity of our approach in enhancing textual information through the extraction of image descriptions using MLLM. 2) Conversely, the multimodal approach performs better than the unimodal approach, underlining the benefit of leveraging information from multimodalities. By fusion and alignment of multimodal information, the model's detection capabilities are significantly enhanced. 3) As illustrated in Table 4, the "BERT-(Text+INTE+IMGDES)" yielded the highest performance, which validates the utility of incorporating intentions derived from social media. Social intentions provide a more comprehensive view of users' psychological states and immediate posting motivations. Therefore, enriching the model with these insights information can significantly enhance its ability to identify sarcastic remarks.

## 6.4 Ablation Study

In this stage, we conducted an ablation experiment to assess the impact of image descriptions and intention information on sarcasm detection tasks. The experimental outcomes, as depicted in Table 4, lead to several insightful observations. First, image description and intentions contribute significantly to sarcasm detection. This is evidenced by the enhanced performance of the BERT-(Text+IMGDES) compared to their counterparts, BERT, which do not incorporate image descriptions. A noteworthy finding is that BERT-(Text+INTE) outperforms BERT-(Text+IMGDES) because the intention is based on further refinement of the original text, image description, and key information, which contains more information that is useful and consistent with human information activities, this information is more helpful for sarcasm detection tasks. Besides, integrating both image description and intention resulted in the most effective result, surpassing the state-of-the-art in multimodal sarcasm detection. This emphasizes the effectiveness of extracting intention information from large-scale models to grasp the user's underlying thoughts, which means that the recognition effect of sarcasm detection data depends on the ability to understand the user's thoughts and motivations accurately.

## 7 Conclusions

In this paper, we introduce MIKO, an innovative framework tailored for acquiring social intention knowledge from multimodal social media posts. Our approach incorporates a hierarchical methodology to extract essential social information and intentions. This process leverages a large language model and well-designed prompts to capture users' posting intentions from social posts effectively. Furthermore, we meticulously annotate the typical scores of selected assertions, enriching them with human knowledge to establish a robust benchmark. We have conducted comprehensive evaluations to validate the effectiveness and utility of the distilled intention knowledge extracted by our framework. In the future, we aim to broaden the scope of MIKO by adapting it to diverse domains, behavioral types, languages, and temporal contexts. This expansion is anticipated to significantly enhance the capabilities of various social media applications such as sentiment analysis.

## Acknowledgments

We thank the anonymous reviewers for their helpful and constructive comments. The authors of this paper were supported by the NSFC through grants U20B2053 and 62302023, the RIF (R6020-19 and R6021-20), and the GRF (16211520 and 16205322) from RGC of Hong Kong. We also thank the support from the UGC Research Matching Grants (RMGS20EG01-D, RMGS20CR11, RMGS20CR12, RMGS20EG19, RMGS20EG21, RMGS23CR05, RMGS23EG08).

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
