# OpenReview forum: "MIKO: Multimodal Intention Knowledge Distillation from Large Language Models for Social-Media Commonsense Discovery"
_acmmm.org/ACMMM/2024/Conference — MM2024 Poster_

### Official Review · Reviewer_8Bao · 2024-05-08

**Rating:** 4
**Confidence:** 2

**Summary:**

This study introduces an innovative multimodal intention generation model. It leverages a Multimodal Large Language Model to extract textual descriptions from images. Subsequently, these descriptions, along with the image text, are fed into a Language Model to identify key information. Another Language Model then integrates all image descriptions, image text and key information to generate the final intention. Human evaluation confirms the high quality of intentions generated by this framework. Additionally, the model is applied to sarcasm detection, demonstrating its efficacy in downstream tasks.

**Strengths:**

1. Novelty: Prior to the ultimate intention generation, this paper suggests a key information extraction stage aimed at filtering out extraneous noise.
2. Adequate evaluation: Distilling a student model serves to illustrate the downstream advantages derived from applying intention knowledge.

**Limitations:**

1. Insufficient evaluation: In terms of intention detection, LLava-v1.5-13B and LLava-v1.6-vicuna-7B exhibited notably superior performance compared to LLama2-7B, even without fine-tuning. Notably, LLava-v1.6-vicuna-7B shares the same size as LLama2-7B. Therefore, the rationale behind choosing LLama2-7B over fine-tuning the similarly sized LLava-v1.6-vicuna-7B warrants exploration.
2. Insufficient evaluation: In the context of sarcasm detection during downstream tasks, one model, HKE, demonstrates performance closely comparable to the proposed model. It would be beneficial to elucidate the reasons behind its superior performance.

**Suitability:**

3

---

### Official Review · Reviewer_TEd2 · 2024-05-20

**Rating:** 4
**Confidence:** 2

**Summary:**

This paper focuses on a novel approach to enhance the understanding and processing of the intention from text and images on social media platforms. The main objective is to distill knowledge from large language models (LLMs) to improve the performance and accuracy of multimodal models in identifying and interpreting user intentions. The authors propose a methodology that leverages the strengths of LLMs to create more efficient and effective multimodal models, which are essential for various applications such as sentiment analysis, content moderation, and personalized recommendations on social media.

**Strengths:**

The topic is very interesting.  The author introduce a unique framework for understanding intention by distilling knowledge from large language models (LLMs) to enhance multimodal models, effectively bridging a significant research gap. Its theoretical approach is robust, aligning text and image representations for better coherence. Technical correctness is ensured through a meticulous distillation process, and the evaluation is comprehensive, showing superior performance in sentiment analysis and user intention prediction.

**Limitations:**

1. The method seems not to be solid. From my opinion, I can only see that the author use the VLLM  and design some prompt to extract image information. It seems like a prompt engineering.

2. About the Intention Distillation, I do not know what you really do when you says ``which combines the original post information, image description information, and key information...'' It is more like a feature engineering, not the Distillation. I tried to find why you call it distillation in method, it is more like you use other LLM's generation to extract features and guide your own model to fine-tune? I think this part should be discussed more.

3. Furthermore, Two-stage Annotation is a human intervention work. I was wondering if there are some real benchmarks that everyone in this area agrees.

**Suitability:**

3

---

### Official Review · Reviewer_p3to · 2024-05-25

**Rating:** 3
**Confidence:** 4

**Summary:**

The authors proposed a new framework, Multimodal Intention Knowledge DistillatiOn (MIKO), that elicits user intentions from user posts. This framework prompts the LMM model to generate an informative image description while using LLM models to generate the key information and the user intentions. Subsequently, the authors conducted a series of experiments to prove the effectiveness of the MIKO framework such as comparing their MIKO framework against other open-source large models and using user intentions to perform sarcasm detection.

**Strengths:**

- [Motivation] The authors are working on a novel problem of distilling user intention from user posts. These user intentions can help end-users and models understand the purpose or objective behind the post.
- [User Intention Benchmark w/ Human Annotations] The authors conducted a relatively large-scale human annotation to construct a user intention benchmark (1,000 posts, five annotations per tweet). Subsequently, the authors conducted a statistic analysis and only retained those that conform to human logic and possess comprehensive intention information, adding them to the benchmark. **This resource, if released, will be useful for the community working on sarcasm detection or intention understanding tasks.
- [Robust Experiment and Analysis] The authors conduct many experiments to evaluate the effectiveness of their MIKO framework and the generated user intentions, such as comparing their MIKO framework against other open-source large models and using user intentions to perform sarcasm detection. However, there are several questions surrounding these experiments (refer to limitations)

**Limitations:**

- [Reproducibility] While the authors mentioned that they use LLaVA and ChatGPT in Sections 4.1 and 4.2, the authors should provide the exact version and the number of parameters for each model. Additionally, the authors did not mention what model they used for the keyword extraction (Section 4.3).
- [Benchmark Other LLM] In Lines 722 - 724, the authors claimed that “…, underscoring *the effectiveness and validity of our extracted intention knowledge in guiding the model’s extraction of intention knowledge accurately.* Firstly, models fine-tuned for specific tasks (in this case, intentions) typically perform better. Hence, saying that LLaMA2-7B (LoRA Fine-Tuned) demonstrates the effectiveness and validity of extracting intention knowledge might not be a very sound claim. To substantiate this claim better, the authors should fine-tune LLaMA2-7B  without the image description and the key information for comparison. Additionally, the performance of LLaVA in a zero-shot inference approach achieves a comparable performance close to LLaMA2-7B. This raises the question of whether instruction fine-tuning LLaVA without the image description and key information will lead to better performance.
- [Marginal Performance w/ No Statistic Testing] Compared to the HKE model, the performance improvement from using BERT-(Text-IMGDES-INTE) is rather marginal. Additionally, the authors did not provide any statistical test on the model performance (e.g., performance aggregated across different runs or p-value testing).

Questions (Potential Limitations)

- [Why use both LMM & LLM] Why does the author use the LMM to describe the image before using LLM for keyword extraction? Is it not possible to use LMM to perform the keyword extraction given the image and text right away? Are there some considerations behind this? Did the authors perform a comparison between using LMM to extract keywords directly and using a combination of LMM and LLM?

Mistakes:

- [Typo] Line 130 - “kowledge” → “knowledge”
- [Typo] Line 148 - “publically” → “publicly”
- [Consistency] Is it -1 for implausible (line 576) or -1 for invisible (line 605)?

**Suitability:**

3

---

### Meta-Review · Area_Chair_DJGb · 2024-06-29

**Recommendation:** Accept (Poster)
**Confidence:** 5

**Metareview:**

The paper addresses important aspects of multimodal AI understanding but needs critical improvements for clarity and rigor. Key concerns include specifying model versions clearly (LLaVA, ChatGPT) and detailing the model used for keyword extraction. The claim regarding LLaMA2-7B's performance enhancement through fine-tuning requires validation without image descriptions and key information supported by statistical tests. Further clarification is needed on distinguishing intention distillation from feature engineering and justifying the two-stage annotation process against established benchmarks. The authors are advised to address these comments in the camera-ready version of the paper.